# Changes in social contacts in England during the COVID-19 pandemic between March 2020 and March 2021 as measured by the CoMix survey: A repeated cross-sectional study

Amy Gimma[1]*, James D. Munday[1], Kerry L. M. Wong[1], Pietro Coletti[2], Kevin van Zandvoort[1], Kiesha Prem[1], CMMID COVID-19 working group[¶], Petra Klepac[1], G. James Rubin[3], Sebastian Funk[1], W. John Edmunds[1], Christopher I. Jarvis[1]

1 Centre for Mathematical Modelling of Infectious Diseases, Department of Infectious Disease Epidemiology, London School of Hygiene & Tropical Medicine, London, United Kingdom, 2 UHasselt, Data Science Institute and I-BioStat, Hasselt, Belgium, 3 Department of Psychological Medicine, King's College London, Denmark Hill, London, United Kingdom

¶ Membership of the CMMID COVID-19 working group is provided in the Acknowledgements.
* Amy.Gimma@lshtm.ac.uk

**Data Availability Statement:** CoMix UK survey data used for this study are available to download

## Abstract

### Background

During the Coronavirus Disease 2019 (COVID-19) pandemic, the United Kingdom government imposed public health policies in England to reduce social contacts in hopes of curbing virus transmission. We conducted a repeated cross-sectional study to measure contact patterns weekly from March 2020 to March 2021 to estimate the impact of these policies, covering 3 national lockdowns interspersed by periods of less restrictive policies.

### Methods and findings

The repeated cross-sectional survey data were collected using online surveys of representative samples of the UK population by age and gender. Survey participants were recruited by the online market research company Ipsos MORI through internet-based banner and social media ads and email campaigns. The participant data used for this analysis are restricted to those who reported living in England. We calculated the mean daily contacts reported using a (clustered) bootstrap and fitted a censored negative binomial model to estimate age-stratified contact matrices and estimate proportional changes to the basic reproduction number under controlled conditions using the change in contacts as a scaling factor. To put the findings in perspective, we discuss contact rates recorded throughout the year in terms of previously recorded rates from the POLYMOD study social contact study.

The survey recorded 101,350 observations from 19,914 participants who reported 466,710 contacts over 53 weeks. We observed changes in social contact patterns in England over time and by participants' age, personal risk factors, and perception of risk. The mean reported contacts for adults 18 to 59 years old ranged between 2.39 (95% confidence interval [CI] 2.20 to 2.60) contacts and 4.93 (95% CI 4.65 to 5.19) contacts during the study

from Zenodo (https://zenodo.org/record/4905746#.YcNN1BPP30o) and the analysis code can be found on Github (https://github.com/amygimma/comix_uk_summary_analysis).

**Funding:** The following funding sources are acknowledged as providing funding for the named authors. This research was partly funded by the Bill & Melinda Gates Foundation (INV-003174: KP, PK). Elrha R2HC/UK FCDO/Wellcome Trust/This research was partly funded by the National Institute for Health Research (NIHR) using UK aid from the UK Government to support global health research. This project has received funding from the European Union's Horizon 2020 research and innovation programme - project EpiPose (101003688: AG, KP, PK, WJE). PC received funding from the European Research Council (ERC) under the European Union's Horizon 2020 research and innovation program (Grant Agreement 682540 TransMID). FCDO/Wellcome Trust (Epidemic Preparedness Coronavirus research programme 221303/Z/20/Z: KvZ). This research was partly funded by the Global Challenges Research Fund (GCRF) project 'RECAP' managed through RCUK and ESRC (ES/P010873/1: CIJ). NIHR (PR-OD-1017-20002: WJE). UK MRC (MC_PC_19065 - Covid 19: Understanding the dynamics and drivers of the COVID-19 epidemic using real-time outbreak analytics: WJE). Wellcome Trust (210758/Z/18/Z: JDM, SFunk). This research was partly funded by the Royal Society under award (RP\EA\180004: KP). The funders had no role in study design, data collection and analysis, decision to publish, or preparation of the manuscript.

**Competing interests:** The authors have declared that no competing interests exist.

**Abbreviations:** CI, confidence interval; COVID-19, Coronavirus Disease 2019; GAM, generalised additive model; IQR, interquartile range; LSHTM, London School of Hygiene & Tropical Medicine; NHS, National Health Service; R, basic reproduction number; $R_c$, basic reproduction number under controlled conditions; $R_t$, reproduction number at time t; REACT Study, Real-time Assessment of Community Transmission Study; SARS-CoV-2, Severe Acute Respiratory Syndrome Coronavirus 2; SD, standard deviation; STROBE, STrengthening the Reporting of OBservational studies in Epidemiology.

period. The mean contacts for school-age children (5 to 17 years old) ranged from 3.07 (95% CI 2.89 to 3.27) to 15.11 (95% CI 13.87 to 16.41). This demonstrates a sustained decrease in social contacts compared to a mean of 11.08 (95% CI 10.54 to 11.57) contacts per participant in all age groups combined as measured by the POLYMOD social contact study in 2005 to 2006. Contacts measured during periods of lockdowns were lower than in periods of eased social restrictions. The use of face coverings outside the home has remained high since the government mandated use in some settings in July 2020. The main limitations of this analysis are the potential for selection bias, as participants are recruited through internet-based campaigns, and recall bias, in which participants may under- or over-report the number of contacts they have made.

## Conclusions

In this study, we observed that recorded contacts reduced dramatically compared to prepandemic levels (as measured in the POLYMOD study), with changes in reported contacts correlated with government interventions throughout the pandemic. Despite easing of restrictions in the summer of 2020, the mean number of reported contacts only returned to about half of that observed prepandemic at its highest recorded level. The CoMix survey provides a unique repeated cross-sectional data set for a full year in England, from the first day of the first lockdown, for use in statistical analyses and mathematical modelling of COVID-19 and other diseases.

## Author summary

### Why was this study done?

- Mathematical models can be used to better understand the transmission dynamics of Coronavirus Disease 2019 (COVID-19) and could be strengthened by empirical evidence of the number of social contacts made under pandemic conditions.

- We identified a need for real-time social contact data to inform outbreak models, as we expected social contact behaviour to change due to perceived risk and in response to government policies restricting social contact over the course of the pandemic.

- We launched the CoMix social contact and behavioural study on March 24, 2020 to capture the changes in social contacts, risk perception, and other behaviours, such as hand hygiene and the use of face coverings.

### What did the researchers do and find?

- During the most stringent lockdown in the UK, we found that the mean number of reported contacts in England was about 75% less than prepandemic measures for adults over the age of 17.

- Throughout the year, the mean number of contacts remained low—only reaching about 50% of the prepandemic levels, even during periods of relatively few policies remaining to restrict social activity.

- During each lockdown, contacts returned to similar levels as the first lockdown for adults, while the mean number of contacts for children depends on whether or not schools were open for in-person learning.

### What do these findings mean?

- Throughout the year, the UK government, which governs England, used the CoMix social contact data to monitor social contacts and as an early indication of changes to the basic reproduction number.

- While these data have been put to use in real time, both researchers and policymakers need to take into account some limitations of the CoMix study results, including the self-reporting of social contacts, which could lead to bias as a result of inaccurate memory or due to social pressure to report more or fewer contacts.

- These data will continue to be used by researchers and policymakers to monitor changes in social behaviour, to model transmission of COVID-19 and other diseases, and to make important policy decisions.

## Introduction

Since early 2020, governments across the world have asked or required people to change their behaviour in an attempt to slow transmission of the Severe Acute Respiratory Syndrome Coronavirus 2 (SARS-CoV-2) virus. In England, the government has implemented a variety of measures over the course of the pandemic, including 3 separate national "lockdowns" [1–5] as well as other local and national measures [6]. In addition, guidance has been issued on risk mitigation measures during social interactions, including meeting outdoors, maintaining space between people, frequent handwashing or use of hand sanitiser, and the use of face coverings (masks).

A key component in the transmission of respiratory viruses is the rate of close social contacts in a population [7]. Social contact studies have been demonstrated as effective in parameterising mathematical disease models to simulate an outbreak [7]. Previous social contact studies have captured contact rates, using a variety of methods including paper diaries, online surveys, and technology capable of recording proximity [8]. Contact matrices are composed of the mean number of contacts between participants of a given age group by the age group of contacts. Contact matrices are used in mathematical modelling to parameterise outbreak simulations and to estimate changes in the basic reproduction numbers. They can also be plotted graphically to visualise the differences in contact patterns by age over time or in different settings, with the age group of participants plotted on one axis, the age group of contacts plotted on the other, and each cell represents the mean number of contacts between the corresponding ages.

Since the Coronavirus Disease 2019 (COVID-19) pandemic began to spread around the world, several contact studies have been conducted to parameterise models during increased risk perception in communities and legal restrictions to social contact [9]. However, no studies, to our knowledge, either during or prior to the COVID-19 pandemic have consistently recorded social contacts in the same population with returning participants over a long period of time or in combination with participant risk perception and behaviours associated with work, travel, education, and leisure.

We conducted weekly cross-sectional surveys to measure the social contacts, behaviours, and attitudes of people in the UK to quantify social interactions over time and have previously described early findings during the first week of lockdown in England (March 24 to 27, 2020) [10]. In this paper, we aim to describe observed contact patterns and behaviour in England based on the CoMix social contact survey collected between March 24, 2020 (the first day of the first national lockdown in the UK) and March 29, 2021 (the final day of the third national lockdown in England). We present descriptive analyses showing the mean number of contacts people reported and how these differed during 3 national lockdowns, periods with more relaxed restrictions, and over the Christmas holiday period. We provide a 1-year detailed account of contact behaviour in England during the first year of the pandemic, create a historical record for future study and policymaking, and improve understanding of the patterns of disease spread and the effectiveness of different policies on reducing contacts to suppress transmission.

While past epidemics have relied on retrospective analyses of the impact of control measures on the course of the epidemic [11], the CoMix study has enabled real-time assessment of the impact of control measures on social contacts, one of the main drivers of transmission. The frequency of our surveys have allowed us to capture social contact behaviour under many different social contact policies and during periods of increasing, plateauing, and decreasing COVID-19–related hospitalisations. This timely study has enabled rapid feedback for policy interventions to the UK government, been used as an early indicator of changes or potential changes to the basic reproduction number [10,12], and provides invaluable data to parameterise models and to inform future study of the transmission and control of respiratory diseases.

## Methods

### Study design

CoMix is an international online behavioural survey that has been running weekly since it launched on the March 24, 2020. While the study is described as a repeated cross-sectional survey, participants were invited to participate in as many as 10 survey rounds, which resulted in the availability of longitudinal data. In the UK, participants are invited to the survey and subsequently asked to respond once every 2 weeks, with 2 panels of participants who respond in alternating weeks. Initially, each panel consisted of roughly 1,500 adult participants, at least 18 years old, increasing to about 2,500 participants each week from August 2020 (Fig 1D). In May 2020, we launched 2 additional panels (each of approximately 500 participants) designed to collect data on children's contact patterns. Parents (at least 18 years old) completed the surveys on behalf of one of their children (<18 years old) who lived in the same household, based on which child had the closest upcoming birthday at the time of the first survey.

A UK representative sample was recruited by the market research company Ipsos MORI using quota sampling, with quotas based on age, gender, and region. Ipsos MORI recruits through a combination of social media, web advertising, and email campaigns and partners with other companies when necessary to meet quotas. Participants agreed to data collection under the lawful basis of research in the public interest and gave informed consent to

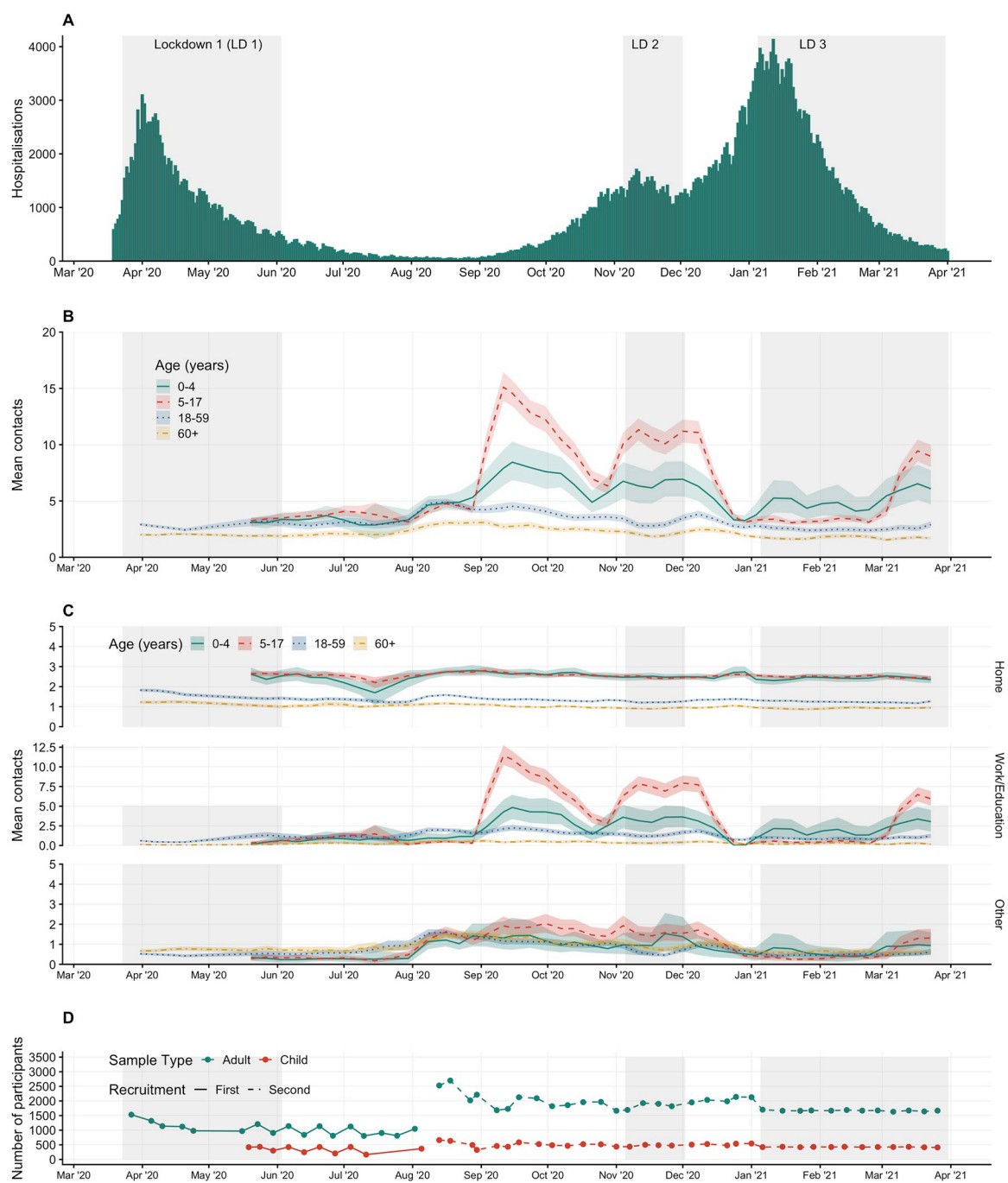

**Fig 1. Mean contacts over time by age and by age and setting with timeline of survey participation with 95% CI of bootstrapped mean. (A)** Hospitalisations due to COVID-19 in England. **(B)** Mean contacts and 95% bootstrapped CIs in adults and children in age groups of 0 to 4, 5 to 17, 18 to 59, and 60 or more year. **(C)** Mean contacts and 95% bootstrapped CIs by age group and setting. **(D)** The number of participants and when they respond by panel over time. CI, confidence interval; COVID-19, Coronavirus Disease 2019; LD, lockdown.

participate in the study. All analyses were carried out on anonymised data. The study and method of informed consent were approved by the ethics committee of the London School of Hygiene & Tropical Medicine (LSHTM; reference number 21795). All participants

demonstrated eligibility by confirming their age, awareness of data privacy protections and the right to decline to participate, and agreement to participate. Participants received compensation for each completed survey and received a bonus for participating in at least 8 waves.

The panels were updated with a fully new sample in August 2020, after the initially planned period of the study was completed, and there was a high turnover of participants throughout January 2021 as participants reached their survey limit or dropped out of the study and new participants were recruited. Participants were included for a maximum of 10 survey rounds in the first group of panels (before August 9) and 8 in the second group (after August 8) to reduce the burden of participating on individuals. Due to different policy implementation among the 4 nations of the UK, in this paper, we restrict analysis to participants who reported living in England.

The survey design is based on the POLYMOD contact survey [13] with additional questions about work and school attendance, household composition, use of public transportation, and a variety of others. Details of the early rounds of the CoMix study including the protocol and survey instrument have been published previously [10], and details of the updated protocol and survey instruments are provided in the Supporting information. The original sample size for the UK was 1,500 adults every 2 weeks, for a total of 16 weeks (8 surveys). With an estimated approximately 12% dropout rate per wave of interviews, this would have resulted in 500 individuals after the final wave. However, we later made the decision to run the survey in 2 panels running on alternating weeks to make data available on a weekly basis, and we topped up the second panel to boost participation. When panels for children's data were initially recruited, we aimed for approximately 500 participants in the first panel, with panel top-ups as needed based on operational analysis needs and recruitment limitations. In August, we recruited 2 fully new panels that included observations for both adult and children's data, with an aim for about 3,000 per week with panel top-ups. This resulted in continued recruitment of participants for the final panels. All changes have been made after approval from LSHTM ethics committee. The protocol for the study outlined a set of aims and objectives and proposed multiple analytical outcomes to be met over the course of several publications; however, as this study was meant to provide real-time data and address changing epidemiological priorities, the proposal did not limit the study to any particular methods. Some of these aims have already been met in prior publications, some will be addressed in this study, and some studies are ongoing. We have added analyses as new study questions and analysis techniques have been identified. This study is reported as per the STrengthening the Reporting of OBservational studies in Epidemiology (STROBE) guideline (S4 Text).

## Survey design

**Reporting of contacts.**   Contacts that occurred on the day prior to the survey were reported in 2 ways: individual contacts and group contacts. First, participants were asked to list each contact and their characteristics separately ("individual contacts"). Second, we asked participants to report the total number of contacts they had at work, school, or other settings for the age groups 0 to 17, 18 to 59, and 60+, both overall and for physical contacts only ("group contacts"). We define direct contact as anyone who met the participant in person with whom at least a few words were exchanged or physical contact was made. Questions on group contacts were included at the end of the survey, and they were added to surveys from the May 14, 2020 to accommodate individuals—such as those working in patient- or public-facing roles —who could not record details of all individual contacts that they made. Since August 2020, participants were also asked to describe the average time spent with group contacts and any transmission-related precautions they implemented, including distancing, wearing face coverings, and performing hand hygiene.

**Demographic information.**   The survey captures information about participant demographics, employment status, and whether participants attended work (or school/university for participants <18 years old and self-reported students) in the previous week and on the day they recorded contacts. We group participants by age into the groups 0 to 4, 5 to 11, 12 to 17, 18 to 29, 30 to 39, 40 to 49, 50 to 59, 60 to 69, and over 70. We group areas into 6 regions: East of England, Greater London, Midlands, North East and Yorkshire, South East, and South West. Participants were asked to report how they describe their gender, with the options of "Female," "Male," "In another way," or "Prefer not to answer." For analyses, due to the small number of participants answering the gender question with "In another way" or "Prefer not to answer," we grouped these with missing values into the category "Other." A socioeconomic category was assigned to participants based on their reported occupation (see S1 and S3 Text for more information).

**Risk perception, status, and mitigation.**   Participants were asked questions about their performance of risk mitigating activities and asked to respond to statements regarding their perception of risk. Participants were asked to respond to the following statements: (i) "I am likely to catch coronavirus"; (ii) "I am worried that I might spread coronavirus to someone who is vulnerable"; and (iii) "Coronavirus would be a serious illness for me" with the Likert scale of "Strongly agree," "Tend to agree," "Neutral," "Tend to disagree," and "Strongly disagree". Participants self-reported whether or not they considered themselves to be high risk based on definitions given in the survey, which changed between survey versions as government advice changed (see questionnaires for details). Participants were also asked whether they wore a face covering and in which settings, whether they had washed their hands in the 3 hours prior to the survey, and whether they sanitised their hands in the 3 hours prior to the survey.

## Analysis time periods

We categorised the dates of contacts in our survey into 9 time periods to compare descriptive statistics and calculate contact matrices. The 9 time periods were selected to reflect 5 stringency levels of nonpharmaceutical interventions we defined as lockdown, lockdown with schools open, lockdown easing, relaxed restrictions (school holiday), and relaxed restrictions (schools open) based on guidance released by the UK government (Tables 1 and 2) [1–5].

Previously published analyses considered the period between September 2 and November 5, 2020 (i.e., when the second lockdown started), when England was under a range of local and

**Table 1. Study periods by intervention type and date range in England.**

| Number | Study period | Stringency | Date range |
|---|---|---|---|
| 1 | Lockdown 1 | Lockdown | March 23 to June 3, 2020 |
| 2 | Lockdown 1 easing | Easing | June 4 to July 29, 2020 |
| 3 | Reduce restrictions | Relaxed restrictions | July 30 to September 3, 2020 |
| 4 | Schools open | Relaxed restrictions with schools open | September 4 to October 26, 2020 |
| 5 | Lockdown 2 | Lockdown with schools open | November 5 to December 2, 2020 |
| 6 | Lockdown 2 easing | Easing | December 3 to December 19, 2020 |
| 7 | Christmas | Relaxed restrictions | December 20, 2020 to January 2, 2021 |
| 8 | Lockdown 3 | Lockdown | January 5 to March 8, 2021 |
| 9 | Lockdown 3 with schools open | Lockdown with schools open | March 9 to March 29, 2021 |

Nine time periods reflect 5 stringency levels of nonpharmaceutical interventions we defined as lockdown, lockdown with schools open, lockdown easing, relaxed restrictions (school holiday), and relaxed restrictions (schools open) that we created based on guidance released by the UK government. Not all survey dates are included in a study period.

**Table 2. Policy interventions in England for study periods.**

| Policy area | Policy type | Lockdown | Easing | Relaxed restrictions | Relaxed restrictions/ schools open | Lockdown/ schools open | Relaxed restrictions[1] (Christmas) |
|---|---|---|---|---|---|---|---|
| Workplace policies | Work from home orders (except for essential workers) | ✓ | ✓ | | | ✓ | ✓[2] |
| | Workplaces open with social distancing restrictions | | ✓ | | | | |
| | Workplace open without social distancing restrictions | | | ✓ | ✓ | | ✓[3] |
| School policies | Schools closed | ✓ | ✓ | | | | |
| | School holidays | | | ✓ | | ✓ | ✓[2] |
| Mask policies | Mandatory masks in some areas | ✓ | ✓ | ✓ | ✓ | ✓ | ✓ |
| Restaurant policies | Restaurants open with restrictions | | ✓ | ✓ | ✓ | | ✓[3] |
| | Restaurants closed | ✓ | | | | ✓ | ✓[2] |
| Other factors | Incentives for dining out | | | ✓ | ✓ | | |
| | Holiday travel restrictions | | | | | | ✓[2] |
| | Study period numbers | 1, 8 | 2, 6 | 3 | 4 | 5, 9 | 7 |

Summary of selected COVID-19–related policies implemented during the 9 study periods, as defined in Table 1.

[1]This period is categorised as a "Relaxed restrictions"; however, this time period also included a period of varied social restrictions for the Christmas holiday.

[2]Policy only effective during the Christmas holiday and may have varied by region.

[3]Policy in place except for the Christmas holiday period.

COVID-19, Coronavirus Disease 2019.

less stringent restrictions, and, therefore, this time period has not been included as a study period for this paper [6].

## Statistical analysis plan

R version 4.0.5 was used for all analyses, and the code and data are available on GitHub (see Data Availability Statement) [14].

**Descriptive.** We calculated summary statistics of the age, gender, socioeconomic status, household size, and National Health Service (NHS) region for participants for each analysis time period and survey panel. While parents answer as proxies for children in the study, we describe the designated child as the "participant" where applicable. We calculated the number and percentage of participants that completed 1, 2 to 3, 4 to 5, and 6 or more rounds of the survey by participant characteristics.

**Mean contacts.** We calculated the mean number of contacts and associated confidence intervals (CIs) with 1,000 samples using clustered bootstrapping [15]. Each participant was sampled with replacement and then all observations for selected participants were included in a bootstrapped sample to account for dependency from repeated observations of the same participants. We calculated the mean number of contacts with a moving window over 2-week, overlapping intervals to increase the sample size per estimate and to include all participants from simultaneously running panels. While the initial panels were recruited to be representative of the UK population, we used poststratification weights of the mean by age group and gender (if available) to address bias introduced by differences between each sample and the UK population [16]. We report contacts by age groups for preschool-age children (<5 years old), school-age children (5 to 17 years old), adults (18 to 59 years old), and the elderly (60+ years old).

Post-stratified weights were assigned by the age groups 0 to 4, 5 to 11, 12 to 17, 18 to 29, in 10 year age bands from 30 to 69 and 70 years old and over. We used the World Population

Prospects 2019 standard projections overall and by gender for the 2020 UK population [17]. Participants with missing ages were not included in this analysis. Estimates of the 2-week intervals are presented with the data points aligned to the central time point of each survey round, and, therefore, each data point shown is derived from information up to 1 week before and after the labelled date. We plotted hospitalised cases of COVID-19 in England alongside mean contact data by age and setting to illustrate the relationship between mean contacts and cases. We used hospitalisation data from the UK government online coronavirus dashboard and filtered for cases only reported in England [18], which we acquired using the *covidregionaldata* R package [19]. We repeated the same process to calculate mean contacts using the POLYMOD data for comparison [13], although we note that CoMix data was censored to 50 contacts per participant based on previous analyses, while the POLYMOD participants in the UK were limited to 30 contacts when completing the survey.

We calculated the mean number of contacts in various settings: home, work and school (all educational establishments, including childcare, nurseries, and universities and colleges), and "other" (mostly leisure and social contacts, including shopping). The mean number of contacts was influenced by a few participants who reported very high numbers of contacts (often in a work context) relative to the rest of the panel. The distribution of reported contacts are right skewed with high variance. The mean number of contacts shown here was calculated by censoring the maximum number of contacts recorded at 50 per individual per day to reduce the variance, meaning we counted any individual who reported more than 50 contacts as if they reported 50 contacts to reduce the weight of individuals reporting high numbers of contacts on the mean. We have found in previous analyses that censoring at 50 contacts most closely reflects changes in contacts relative to changes in reproduction number at time t ($R_t$) over time as estimated by the Real-time Assessment of Community Transmission (REACT) study, a large home testing study conducted in the UK with the aim of quantifying COVID-19 transmission and infections [20,21].

We report bootstrapped mean contacts using the method previously described by responses to questions about reported risk and risk perception and by employment and income categories. For Likert style questions, we group participant responses of "Tend to agree" and "Strongly agree" into one category of "Agree," and we group the responses of "Tend to disagree" and "Strongly disagree" into one category of "Disagree." Only adult participants are included in these analyses. For contacts by employment, we only include participants who recorded working on the day in which they were reporting contacts. Participants who declined to answer these questions were not included in the analysis.

**Study periods.** We calculated relative differences in mean contacts between study periods using an individual-level generalised additive model (GAM) [22,23]. We assumed reported contacts followed a negative binomial distribution, modelled using a log link function, with a random effect for participants by age group (0 to 4, 5 to 17, 18 to 59, and 60 years and over) with poststratification weights for age and gender (when available) based on the UK population. We similarly calculate the relative difference in mean contacts between age groups within each study period using a similar GAM function, with only the study period and age group variables exchanged.

**Face coverings.** We present the bootstrapped 95% CIs of the proportions of participants who reported wearing a face covering in any setting for all participants and separately for only those participants who reported contacts outside the household on the day of the survey. Participants who declined to answer these questions were not included in the analysis.

**Contact matrices.** We constructed age-stratified contact matrices for 9 age groups (0 to 4, 5 to 11, 12 to 17, 18 to 29, 30 to 39, 40 to 49, 50 to 59, 60 to 69, and 70+ years old). For child participants and contacts, we did not record exact ages and therefore sampled from the

reported age group with a weighting consistent with the age distribution of contacts for the participants' own age group, according to the POLYMOD survey methods [13]. We fitted a negative binomial model censored to 50 per matrix cell, due to dispersion of the reported number of contacts, to calculate mean contacts between each participant and contact age groups. To find the population normalised reciprocal contact matrix, we first multiplied the columns of the matrix by the mean normalised proportion of the UK population in each age group [13,24]. Then we took the cross-diagonal mean of each element of the contact matrix. Finally, we divided the resulting symmetrical matrix by the population mean normalised proportion of the UK population in each age group.

We used this approach to construct a contact matrix for each of the analysis periods by filtering the contact data by date. For each time period (Table 1), we calculated the dominant eigenvalue of the infectiousness and susceptibility corrected contact matrix ($C_{SI}$), calculated from the measured contact matrix $C_t$, and assumed age-dependent relative susceptibility and infectiousness vectors s and i:

$$C_{SI} = C_t \circ (i \otimes s)$$

We calculated the relative difference in the basic reproduction number under controlled conditions to Lockdown 1 by taking the ratio of the dominant eigenvalue of the effective contact matrices from the period in question and the dominant eigenvalue from Lockdown 1 [7,10,25]. The value of $R_c$, the basic reproduction number under controlled conditions, is defined as the expected number of secondary cases resulting from an initial infection in a completely susceptible population adjusted to changing conditions, in this case the change in social contacts over time. This value does not account for immunity in the population and will therefore be higher than the actual reproduction number at a given time.

$$\Delta R = \frac{C_t \circ (i \otimes s)}{C_{LD1} \circ (i \otimes s)}$$

We applied 2 assumptions of age-dependent susceptibility and infectiousness. First, we assumed that all age groups are equally infectious and susceptible. Second, we applied a weight for relative susceptibility and infectiousness by age as estimated by Davies and colleagues [26] (S1–S4 Text).

## Results

### Participants characteristics

Overall, we recorded 101,350 observations from 19,914 participants who reported 466,710 contacts over 53 weeks (March 23, 2020 to March 29, 2021). About a quarter of the participants ($n$ = 4,574) were proxy respondents (i.e., the survey was completed by parents on behalf of children), and 15,340 were adults. The median number of responses per participant was 6 (min–max 1 to 9) with 20.6% (4,098) responding only once. We did not follow up with participants to gather information about reasons for dropping out.

The sample consisted of 8,714 (52.8%) females and 7,790 (47.2%) males. Participants were assigned socioeconomic category based on occupation by the Ipsos MORI company (see key for socioeconomic categorisation in S1–S4 Text), which categorised 11,743 (63.1%) participants in socioeconomic category A, B, or C1 and 6,880 (36.9%) in C2, D, or E (S2 Fig). The NHS England region with the most participants was the Midlands with 4,029 (20.2%) participants, and the North West had the fewest with 1,931 (9.7%). The characteristics of the participants were consistent over the different analysis periods, with slight variations over the course of the study, particularly in gender balance and household size (Table 3). For instance, around

**Table 3. Participant characteristics.**

| Group | Value | Initial recruitment | Second recruitment | New year |
|---|---|---|---|---|
| Dates | Start | March 23, 2020 | August 9, 2020 | January 2, 2021 |
| | End | August 8, 2020 | January 1, 2021 | March 29, 2021 |
| All | - | 5,080 | 13,087 | 8,455 |
| Adult | - | 3,815 | 10,230 | 6,389 |
| Child | - | 1,265 | 2,857 | 2,066 |
| Age group (children) | 0 to 4 | 190 (15.3%) | 451 (16.3%) | 311 (15.8%) |
| | 5 to 11 | 469 (37.9%) | 1,076 (38.8%) | 749 (38.0%) |
| | 12 to 17 | 580 (46.8%) | 1,245 (44.9%) | 909 (46.2%) |
| | Unknown age* | 26 | 85 | 97 |
| Age group (adults) | 18 to 29 | 556 (14.6%) | 1,872 (18.3%) | 1,070 (16.7%) |
| | 30 to 39 | 602 (15.8%) | 1,786 (17.5%) | 1,167 (18.3%) |
| | 40 to 49 | 653 (17.1%) | 1,624 (15.9%) | 1,069 (16.7%) |
| | 50 to 59 | 722 (18.9%) | 1,890 (18.5%) | 1,183 (18.5%) |
| | 60 to 69 | 708 (18.6%) | 1,826 (17.8%) | 1,147 (18.0%) |
| | 70+ | 574 (15.0%) | 1,232 (12.0%) | 753 (11.8%) |
| Gender | Female | 2,580 (50.8%) | 6,864 (52.4%) | 4,475 (52.9%) |
| | Male | 2,477 (48.8%) | 6,162 (47.1%) | 3,937 (46.6%) |
| | Other | 23 | 61 | 43 |
| Household size | 1 | 698 (13.7%) | 2,276 (17.4%) | 1,398 (16.5%) |
| | 2 | 1,357 (26.7%) | 4,604 (35.2%) | 2,900 (34.3%) |
| | 3 to 5 | 2,769 (54.5%) | 5,894 (45.0%) | 3,968 (46.9%) |
| | 6+ | 256 (5.0%) | 313 (2.4%) | 189 (2.2%) |
| Social group | A—Upper middle class | 247 (4.9%) | 651 (5.0%) | 420 (5.0%) |
| | B—Middle class | 1,278 (25.2%) | 3,423 (26.2%) | 2,305 (27.3%) |
| | C1—Lower middle class | 1,535 (30.2%) | 4,357 (33.3%) | 2,816 (33.3%) |
| | C2—Skilled working class | 967 (19.0%) | 1,979 (15.1%) | 1,297 (15.3%) |
| | D—Working class | 716 (14.1%) | 1,870 (14.3%) | 1,141 (13.5%) |
| | E—Lower level of subsistence | 337 (6.6%) | 807 (6.2%) | 476 (5.6%) |
| NHS region | East of England | 552 (10.9%) | 1,511 (11.5%) | 1,009 (11.9%) |
| | Greater London | 774 (15.2%) | 2,001 (15.3%) | 1,298 (15.4%) |
| | Midlands | 1,001 (19.7%) | 2,688 (20.5%) | 1,751 (20.7%) |
| | North East and Yorkshire | 728 (14.3%) | 2,046 (15.6%) | 1,315 (15.6%) |
| | North West | 630 (12.4%) | 1,093 (8.4%) | 753 (8.9%) |
| | South East | 802 (15.8%) | 2,263 (17.3%) | 1,438 (17.0%) |
| | South West | 593 (11.7%) | 1,485 (11.3%) | 891 (10.5%) |

The number and percentage of participants surveyed during the first 2 panels (Initial recruitment), the beginning of the next 2 panels (Second recruitment), and the period since the end of the Christmas study period (New year), as most of the sample was refreshed by this point. Number of participants presented overall and by sample type, age, gender, household size, social group, and NHS region. Participants are counted once per study period but may have participated in several waves within a study period. Adult and child samples were recruited separately, and percentages of age groups were calculated by sample type. The "Other" gender category includes participants who do not describe themselves as either male or female and those who declined to answer.

*Some parent participants may have incorrectly completed this question. We have included the observation in this dataset and record the ages as "unknown."

NHS, National Health Service.

14% of the participants lived in a single person household in the initial recruitment round versus around 16 to 17% for later recruitment periods.

While participants were recruited to fill quotas by age and gender, participation varies by wave. A total of 32.0% of participants 18 to 29 completed 6 or more rounds of the survey,

while 27.9% completed only 1 round (S1 Table). Moreover, 60 to 69 year olds had the highest percentage of participants complete 6 or more rounds at 64.8% and the lowest percentage of participants completing only 1 round at 10.0%. In children's panels 36.6% to 38.7% of participants in the child's age group completed 6 or more rounds, and 18.9% to 22.5% completed only 1 round, not including those with an unknown age group (S2 Table).

## Mean contacts, risk perception, and face coverings

Overall, mean daily contacts for working-age adults (18 to 59 years) recorded over the study period, weighted by age, gender, and week day of data collection, varied from 2.39 (95% CI 2.20 to 2.60) during periods of lockdown to 4.93 (95% CI 4.65 to 5.19) during the summer of 2020, when many restrictions were relaxed (S3 Table). The adjusted bootstrapped mean contacts for participants in the POLYMOD study 18 to 59 was 11.41 (95% CI 10.75 to 12.08) (S4 Table). Contacts for older adults (60+ years) were consistently lower throughout the study period ranging from 1.55 (95% CI 1.42 to 1.69) to 3.09 (95% CI 2.82 to 3.39) contacts per person per day, while the reported mean for the same ages in the POLYMOD study was 8.19 (95% CI 6.92 to 9.43). Mean recorded contacts for school-age children were more variable, between 2.87 (95% CI 1.59 to 4.74) contacts per day for 0 to 4 year olds during lockdown when their schools were fully or partially closed and 15.11 (95% CI 13.87 to 16.41) contacts per day for 5 to 17 year olds in September 2020 when schools were open (Fig 1B, S3 Table). The mean number of contacts for children 0 to 4 years in the POLYMOD study was 9.01 (95% CI 7.82 to 10.29) and 15.44 (95% CI 14.36 to 16.57) for children 5 to 17 years. Baseline surveys, conducted before the COVID-19 pandemic, give an indication of normal levels of contact. The more recent BBC Pandemic social contact study in 2017 to 2018 had similar results to the POLYMOD study, reporting a mean of 10.5 contacts for all ages [24].

Following the lifting of Lockdown 1 from late May to early July 2020, recorded contacts remained low until August 2020 (Fig 1B). Contact patterns rebounded much more quickly after the second lockdown in December 2020, despite the continuing imposition of restrictions (a tiered system of restrictions was in place in England, which was strengthened after the second lockdown). Reported contacts were very low during the Christmas period, with a modest easing of restrictions over the holiday period in some parts of England and tighter restrictions in others. Finally, adult contact rates remained low during the third lockdown, with substantial restrictions remaining in place through the end of the study period in March 2021. The pattern of schools opening and closing was the main determinant of children's contacts (Fig 1B and 1C).

**Contacts by setting.** For adults, contacts made at home mostly reflected household size (S1 Fig) and were consistently below a mean of 2 contacts per day over the study period, with little change in reported contacts across each of the analysis time periods (Fig 1C). Work and other contacts followed a similar pattern to adults: staying low but steadily increasing towards the end of the Lockdown 1, increasing in August 2020, decreasing slightly, and then returning to levels similar to the Lockdown 1 during the Lockdown 2 in November, and then reducing again over Christmas and throughout Lockdown 3.

During the first lockdown, schools were closed to all except vulnerable children and the children of essential workers, and recorded children's contact rates were very low (Fig 1B and 1C). From early June 2020 until the third week of July 2020 (when schools were closed for the summer vacation), there was a limited reopening of schools, but most parents reported that their children continued to be educated from home. Average recorded contact patterns among children remained very low during this period (Fig 1B and 1C). When schools reopened fully in September 2020, the number of contacts rapidly increased for both school-age (5 to 17) and

preschool-age children (0 to 4), although the increase in contacts in the latter age group was smaller. Children's contacts declined significantly during the "half-term" vacation at the end of October 2020 but remained high during the second national lockdown (November 2020) as schools remained open. Schools were closed for the Christmas period, remained closed during the third national lockdown, and reopened on March 8, 2021. However, preschools were the first educational setting to reopen during the relaxation of the first lockdown and were not closed during the third lockdown. The contact patterns of 0 to 4 year olds reflect this, with mean rates of contact for this age group being higher than other children during the periods when preschools were open but primary and secondary schools were closed.

**Contacts by study period.** We compared the relative difference of mean contacts using a GAM with Lockdown 1 as the reference period, as this was the beginning of the survey and the period of most stringent lockdown measures. The dates for lockdown periods can be found in Table 1. Contacts remained at similar levels to Lockdown 1 (the reference period) through Lockdown 1 easing for all age groups until the Reduced restrictions study period (Fig 2, Table 4). The relative difference was highest for adults over the 12-month study period during the period of Reduced restrictions with the relative difference of 1.59 (95% CI 1.54 to 1.64) for adults ages 18 to 59 years and 1.51 (95% CI 1.45 to 1.57) for adults aged over 60 years. For children, the relative difference was highest while schools were open during the Schools open,

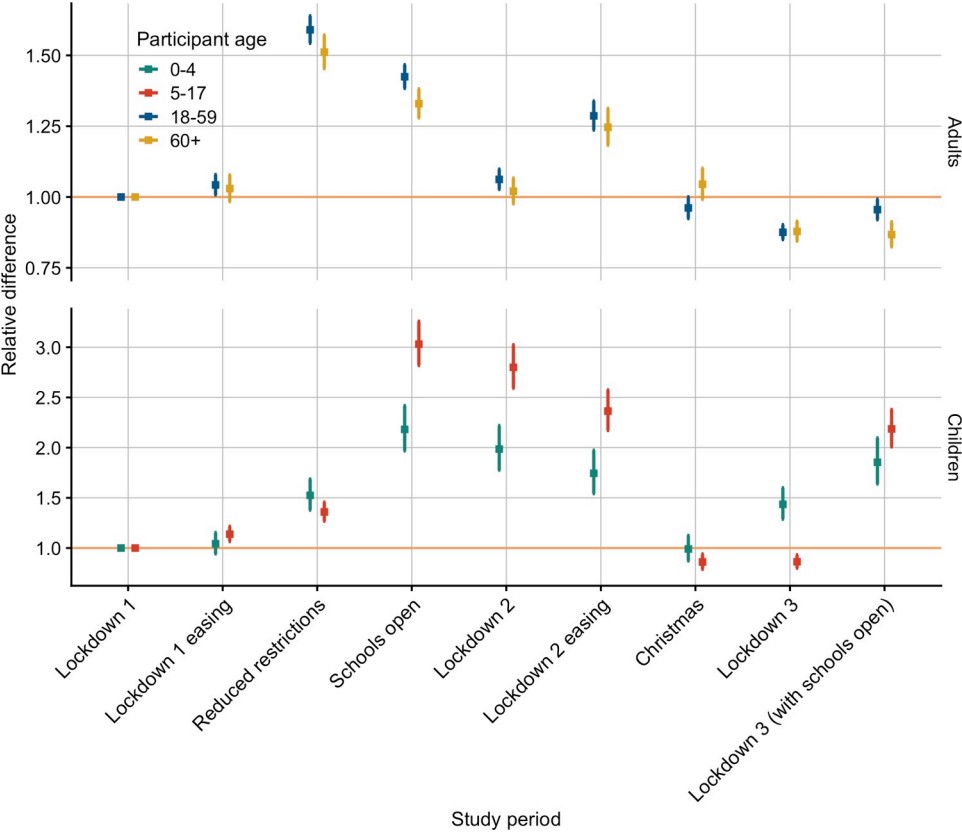

**Fig 2. Relative difference in mean contacts by study period and age group with 95% CIs.** Relative differences calculated using a GAM with Lockdown 1 as the reference period for each age group adjusted to the UK population by age and gender (when available) for the age groups 0 to 4, 5 to 17, 18 to 59, and over 60 years old. Note the facets have different scales on the y-axes. Table 1 provides corresponding dates for each study period. CI, confidence interval; GAM, generalised additive model.

**Table 4. Relative difference in mean contacts by study period with 95% CIs.**

| | | Relative difference in mean contacts (95% CI) | | | |
|---|---|---|---|---|---|
| Number | Study period | Ages 0 to 4 years | Ages 18 to 59 years | Ages 5 to 17 years | Ages 60+ years |
| 1 | Lockdown 1 | Ref | Ref | Ref | Ref |
| 2 | Lockdown 1 easing | 1.04 (0.94 to 1.16) | 1.04 (1.01 to 1.08) | 1.14 (1.06 to 1.22) | 1.03 (0.98 to 1.08) |
| 3 | Reduced restrictions | 1.52 (1.38 to 1.69) | 1.59 (1.54 to 1.64) | 1.36 (1.27 to 1.46) | 1.51 (1.45 to 1.57) |
| 4 | Schools open | 2.18 (1.97 to 2.42) | 1.42 (1.38 to 1.47) | 3.03 (2.82 to 3.26) | 1.33 (1.28 to 1.38) |
| 5 | Lockdown 2 | 1.99 (1.78 to 2.22) | 1.06 (1.03 to 1.10) | 2.80 (2.59 to 3.02) | 1.02 (0.98 to 1.07) |
| 6 | Lockdown 2 easing | 1.74 (1.54 to 1.97) | 1.29 (1.24 to 1.34) | 2.36 (2.17 to 2.57) | 1.25 (1.18 to 1.31) |
| 7 | Christmas | 0.99 (0.87 to 1.13) | 0.96 (0.92 to 1.00) | 0.86 (0.79 to 0.94) | 1.05 (0.99 to 1.10) |
| 8 | Lockdown 3 | 1.44 (1.29 to 1.60) | 0.88 (0.85 to 0.90) | 0.86 (0.80 to 0.93) | 0.88 (0.84 to 0.91) |
| 9 | Lockdown 3 (with schools open) | 1.85 (1.64 to 2.10) | 0.96 (0.92 to 0.99) | 2.19 (2.01 to 2.38) | 0.87 (0.82 to 0.91) |

Relative differences calculated using a GAM with Lockdown 1 as the reference period for each age group adjusted to the UK population by age and gender (when available) for the age groups 0 to 4, 5 to 17, 18 to 59, and over 60 years old.

CI, confidence interval; GAM, generalised additive model.

Lockdown 2, Lockdown 2 easing, and Lockdown 3 with schools open study periods, ranging from 1.74 (95% CI 1.54 to 1.97) for ages 0 to 4 to 3.03 (95% CI 2.82 to 3.26) for ages 5 to 17.

Additionally, we compared the relative difference of mean contacts using a GAM with age 18 to 39 years old as the reference age group to compare the mean number of contacts between age groups within each study period, using the age groups 0 to 4, 5 to 17, 40 to 59, and 60 + years as the comparison groups. We chose 18 to 39 years as it was the midrange age group, which could then be compared to children and to older adults. The greatest relative difference of the mean number of contacts compared to 18 to 39 year olds was during Lockdown 2 for ages 5 to 17 with a relative difference of 3.67 (95% CI 3.48 to 3.87), while older adults had a relative difference of 0.67 (95% CI 0.64 to 0.70) (S4 and S5 Figs). The relative difference in contacts was lower during periods of school closure than when schools were open.

**Precautionary behaviours and risk perception.** The majority (around 50%) of participants answered "Neutral" to a statement indicating that they were likely to catch coronavirus, and this remained fairly consistent over the course of the study (S3 Fig), among all adult age groups. Survey participants who agreed with a statement that they were likely to catch coronavirus recorded higher mean contacts (Fig 3A), especially in August 2020 and during the period following the second lockdown. Mean contacts for those who disagreed or were neutral were very similar. Participants who agreed with a statement indicating that they were worried that they might spread coronavirus to others generally had a higher mean number of contacts between the first and second lockdowns than those who disagreed with the same statement (Fig 3B). For example, those age 18 to 59 years who felt they were likely to spread COVID-19 reaching the mean number of contacts 6.20 (95% CI 5.27 to 7.07) in mid-September 2020, while those who disagreed during that time period was 3.47 (95% CI 3.00 to 4.01). In the same time period (mid-September), the mean number of contacts for those who reported that they did not worry that they would spread COVID-19 to someone who is vulnerable was 3.50 (95% CI 2.97 to 4.06) compared to those who reported they were worried 4.96 (95% CI 4.51 to 5.43). During all 3 lockdowns, the mean contact CIs overlap for participants in all 3 categories (Agree, Neutral, and Disagree) responding to this question.

Survey participants aged 18 to 59 years who disagreed that coronavirus would be serious for them reported slightly higher contacts than those who agreed with the statement, while

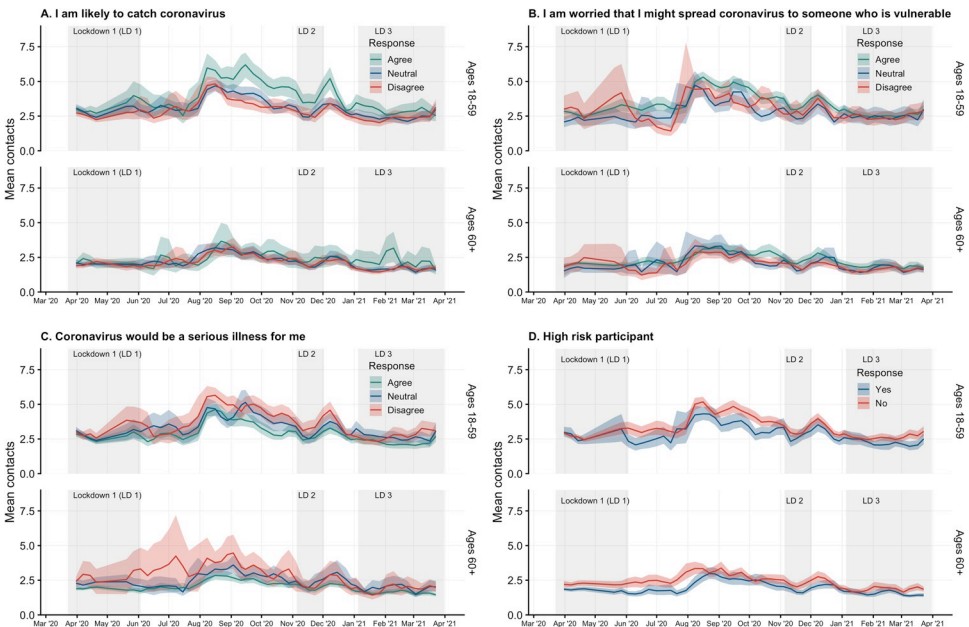

**Fig 3. Mean contacts by risk perception or risk category by adult age groups of 18 to 59 and 60 or more years with 95% CI of bootstrapped mean weighted by age, gender, and weekday.** Participants answered a series of questions about their risk perception with Likert scale response options. Answers of "Strongly agree" and "Somewhat agree" were combined into a category of "Agree, as were answers of "Strongly disagree" and "Somewhat disagree" to "Disagree." (**A**) Answers to the statement "I am likely to catch coronavirus." (**B**) Answers to the statement "I am worried I might spread coronavirus to someone who is vulnerable." (**C**) Answers to the statement "Coronavirus would be a serious illness for me." (**D**) Participant reported they were an individual at high risk for complications as defined in the questionnaire. CI, confidence interval; LD, lockdown.

participants over 60 years of age who disagreed were few in number and reported a wide range of contact behaviours (Fig 3C). Participants who were not high risk generally reported more contacts on average than those who were high risk in both age groups, especially during periods outside of lockdown and towards the end of the third lockdown, with the differences being more pronounced in the over 60 age group (Fig 3D). The highest mean number of contacts for those who did not report that COVID-19 would be a serious illness for them was 5.65 (95% CI 5.12 to 6.20) in mid-August 2020 compared to 4.62 (95% CI 4.20 to 5.04) for those who did agree that COVID-19 would be a serious illness for them in the same time period.

In terms of protective behaviour, the reported use of facemasks at least once on the previous day was low (approximately 12% for 18 to 59 year olds and approximately 3% for 60+ years olds) at the end of March 2020 for participants who reported contacts outside of the household (Fig 4). The proportion who self-reported wearing masks increased gradually for both age groups through June 2020, and a sharp increase in mask use was reported in late July and early August 2020, shortly after mask wearing became mandatory for entering shops on July 24, 2020 [27]. From August 1 through March 26, 2021, mask wearing ranged between 73% and 86% for adults 18 to 59 and between 70% and 84% for adults over 60 among participants with contacts outside the home.

**Employment and income.** Participants who were employed part time consistently reported more contacts on average than full-time or self-employed participants with a wider range of contacts, and full-time workers reported contact means similar or slightly higher than self-employed workers in between lockdowns (Fig 5A).

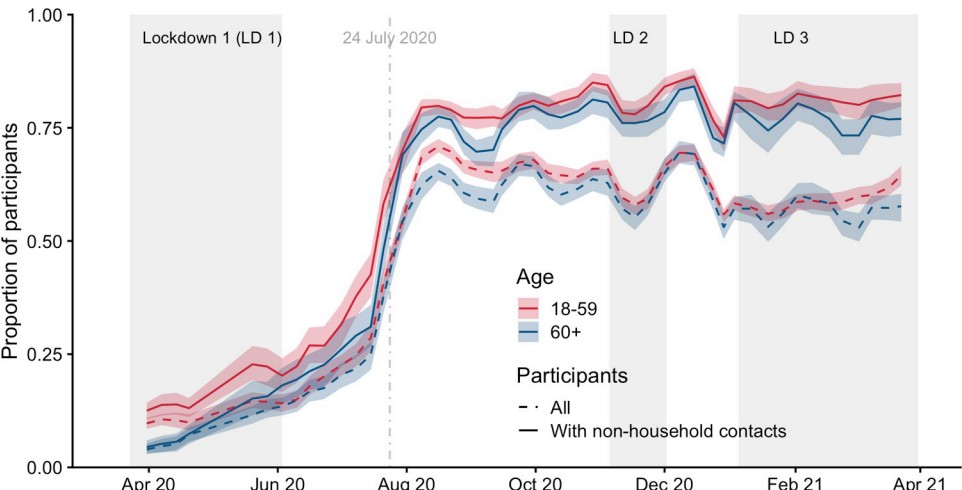

**Fig 4. Proportion of adult participants who report wearing a mask by age category with 95% CI of bootstrapped proportion.** Proportions plotted for all participants and for participants who reported any nonhousehold contacts, with the start date of face covering mandates in some settings indicated on July 24, 2020. CI, confidence interval; LD, lockdown.

Mean contacts for adult participants over 18 years of age grouped by annual income levels (less than £20k, £20k to £44.9k, and over £45k), were similar and follow consistent patterns of decreasing during lockdowns and increasing slightly between the first and second lockdown (Fig 5B).

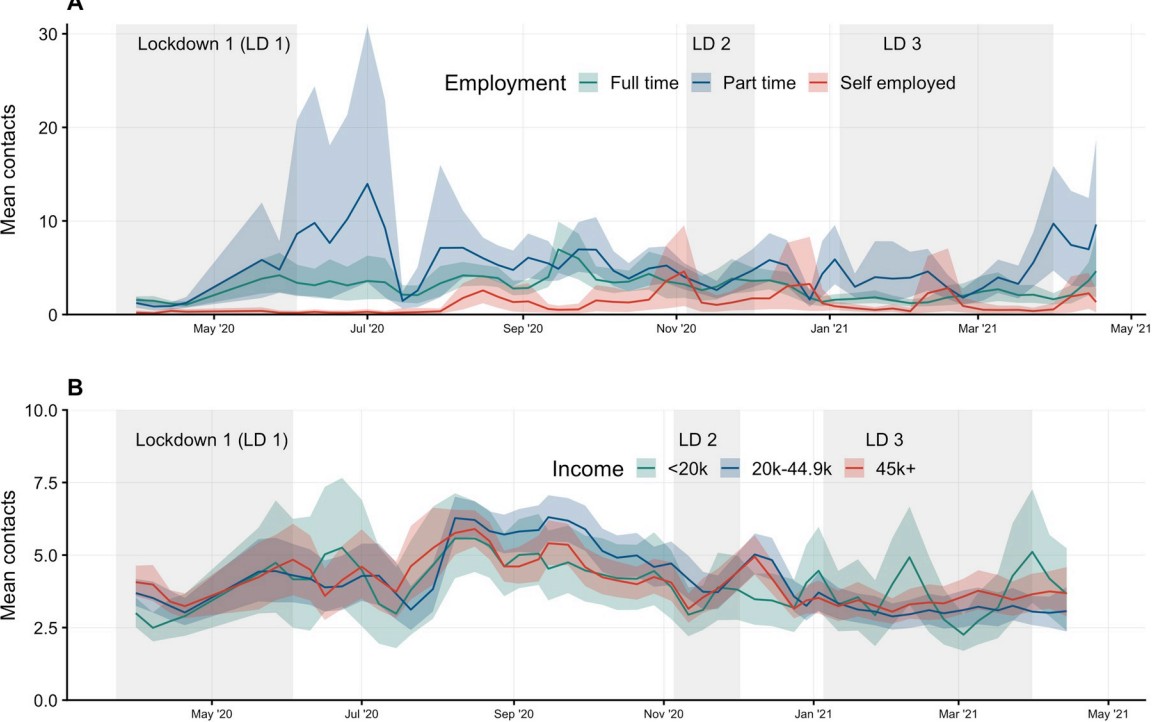

**Fig 5. Mean contacts by employment and income status.** Mean contacts of participants who worked on the previous day and their workplace was open on the previous day weighted by age, gender, and weekday. **(A)** By employment type: full time, part time, or self-employed. **(B)** By annual income level: less than £20k, £20k to £44.9k, and over £45k. LD, lockdown.

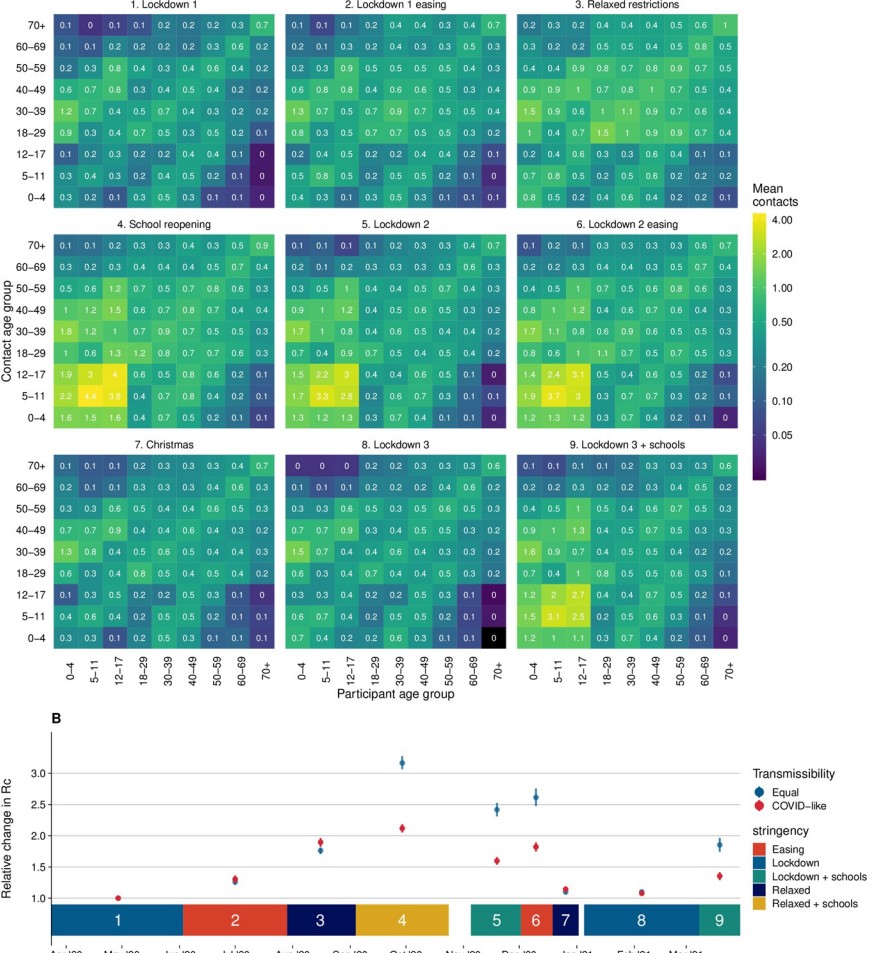

**Fig 6. Contact matrices and their dominant eigenvalues for England in each period considered. (A)** Contact matrices for England across the 9 periods (1. Lockdown 1, 2. Lockdown 1 easing, 3. Relaxed restrictions, 4. School reopening, 5. Lockdown 2, 6. Lockdown 2 easing, 7. Christmas, 8. Lockdown 3, and 9. Lockdown 3 + schools). **(B)** Points show relative change in R_0 (compared to Lockdown 1) based on the dominant eigenvalues of effective contact matrices calculated for periods 1 to 9, with equal transmissibility in all age groups and age-stratified transmissibility based on Davies and colleagues for SARS-CoV-2. Coloured blocks show durations of each period as annotated. COVID, Coronavirus Disease; $R_c$, basic reproduction number under controlled conditions; SARS-CoV-2, Severe Acute Respiratory Syndrome Coronavirus 2.

## Contact matrices

The contact matrices showed overall higher contacts between all age groups in every period compared to Lockdown 1, with increased clustering around the diagonal matrix elements indicating higher rates of contact between those of similar ages (Fig 6). This resulted in higher estimates of the basic reproduction number under controlled conditions ($R_c$). The periods with the highest $R_c$ were between July and August 2020, which corresponds to lockdown easing and government incentives encouraging the public to dine in restaurants, where contacts particularly increased in 18 to 49 year olds and older adults (>60 year olds). As schools reopened in September 2020 following the summer vacation, an increase in contacts between children increased $R_c$ by a factor of 3.17 (95% CI 3.06 to 3.27) relative to Lockdown 1 and 2.12 (95% CI 2.05 to 2.18) when assuming equal transmissibility in all age groups or assuming age-dependent transmissibility estimates relevant to SARS-CoV-2, respectively.

Severe restrictions remained in place over the Christmas holidays (for much of the population) and during Lockdown 3. However, because early childhood education institutions remained open during Lockdown 3, higher contact rates were reported between <4 year olds (S6 Fig). An increase in contacts between school-age children and a slight increase in contacts among adults were found during the periods when schools reopened during Lockdowns 2 and 3 (S6 Fig). This resulted in an $R_c$ much greater than other lockdown periods, for example, during Lockdown 2, we estimated $R_c$ to be 2.42 (95% CI 2.31 to 2.53) and 1.60 (95% CI 1.54 to 1.66) times higher than Lockdown 1, for equal and COVID-like transmissibility by age, respectively.

## Discussion

We conducted a large, detailed repeated cross-sectional social contact survey that has quantified temporal changes in contacts from a representative sample of the English population over the first year of the COVID-19 pandemic. This period (from late March 2020 to late March 2021) encapsulates 3 periods of national lockdowns interspersed with periods in which fewer restrictions were in place. Mean contact rates have remained lower throughout the year, compared to prepandemic contact studies conducted using similar questionnaires [13,24]. Even at the period of minimum restrictions, they were only about half of that observed in the prepandemic surveys. These large reductions in contacts have helped reduce the reproduction number, although only during periods of lockdown has the reproduction number been maintained below 1 [6,21,28].

The survey results suggest that government action was associated with a changes in the mean number of social contacts, with markedly lower contact rates recorded during every lockdown. However, it was not the only plausible determinant. Age was clearly associated with contacts, with children reporting the greatest average number of contacts during periods of school opening. Among adults, younger individuals (18 to 30 years) reported the highest mean numbers of contacts throughout the year and the elderly the fewest—a pattern that is consistent with prepandemic data, albeit at much lower levels of contact. In addition, there appeared to be some association with actual or perceived risk: those who were not in an elevated risk group or did not perceive coronavirus as likely to be severe for them reported higher rates of contact. Employment status was also associated with contact patterns, with part-time workers documenting more contacts than others. Income levels were not strongly associated with mean contacts, nor is there any evidence of seasonality in contacts—the observed temporal changes in contacts corresponded to government action and advice. Although we observed an increase in contacts when restrictions were eased, the change did not occur immediately after the policies were implemented. In particular, the easing of the first lockdown was not associated with a rapid rise in the mean number of contacts until August when the government introduced an incentive scheme to encourage individuals to dine in restaurants, cafes, and bars.

The use of face coverings was also strongly associated with changes in government policy. Although the proportion of individuals reporting having worn a face covering increased gradually over time, the rates of mask wearing increased when it became mandatory for entry into shops on July 24, 2020.

### CoMix in context

The CoMix survey contributes to the growing study of social contacts and their implication in disease transmission. In addition to data collection in the UK, the CoMix study has collected data from participants in 20 additional countries throughout Europe [29]. While social contact

surveys in conducted in various countries demonstrate different social contact patterns and, therefore, should be interpreted with caution when results are applied to a setting other than that of the study setting, our results demonstrate the scale of changes that can occur within a population over time, especially during a pandemic.

Studies prior to the COVID-19 pandemic provide a baseline of contacts in several countries which have been used to project estimates of contacts in other countries [8,30]. Many surveys implemented during the pandemic, conducted in countries throughout the world, provide data on behaviour and social contacts during periods of heightened risk of transmission and with restrictions on social behaviour and have been summarised in a recent review of the literature [9]. To the best of our knowledge, CoMix is the largest repeating cross-sectional survey, which also includes longitudinal data, in comparison to other surveys in the review and appears to be the only survey to have recorded data every week for at least a year. All surveys in the review reflect fewer social contacts during periods of social restrictions throughout 2020 and 2021. A number of mobility indices such as those created by Google and Facebook have been made available during the pandemic, which provide an indication of movement based on monitoring mobile phones and have been used as an indication of changes in behaviour throughout the pandemic [31–33]. However, these are less direct measures that reflect less epidemiologically relevant contacts, and, although previous work has suggested that Google mobility data correlates well with the CoMix data [34], the data are usually shared at aggregate level and, therefore, are impossible to analyse by factors such as age and working status.

## Limitations

The survey is conducted online, using a quota-based sample of individuals who have agreed to participate in marketing surveys. This recruitment method is biased towards people with access to the internet and who may be reached by banner ads, email campaigns, and social media advertisements. Participants only received guidance through the text in the questionnaires and may interpret questions differently. This may be especially evident in the reporting of group contacts. Responses are also subject to recall bias, which may under- or overestimate contacts depending on the nature of the contacts. Additionally, due to child protection concerns and age-dependent ability to complete the survey, children's contacts are collected through a parent acting as a proxy for a child, which may lead to inaccurate reporting. Mean contacts are sensitive to a few participants who report many contacts, which we have addressed by assigning all reports of over 50 contacts to 50 contacts. Further research is needed to create standardised methods for analysing highly dispersed contact data, although a standardised approach may not be feasible as it may be context dependent. Some caution should be taken in comparing the CoMix survey to other contact surveys due to differences in the questionnaires and in survey implementation.

## Conclusions

This study quantifies changes in epidemiologically relevant contact behaviour for 1 full year of the COVID-19 pandemic in England. Contacts have remained suppressed far below normal levels throughout the year, although changes in contacts have occurred following relaxation or tightening of social distancing measures. The CoMix survey is unique in both length and frequency of the data and in its longitudinal study design, which provides a detailed historical record of social behaviour during the COVID-19 pandemic. Importantly, CoMix contact data are age stratified for both participants and contacts and can be used to construct social contact matrices for age-stratified modelling. These data can be used to inform future outbreak response and can be applied to transmission of other infectious diseases, particularly for a large-scale pandemic.

## Supporting information

**S1 Fig. Mean number of contacts by age group and household size.** Bootstrapped means weighted by age, gender, and weekday. Households of 6 or more for the 60 and older age group was omitted due to a low number of participants in the category.
(TIF)

**S2 Fig. Mean contacts and 95% CIs by age group and socioeconomic groups.** Bootstrapped mean contacts of participants weighted by age, gender, and weekday. Social groups ABC1 include the socioeconomic categories A, B, and C1 and the social groups C2DE include the socioeconomic categories C2, D, and E as shown by occupation (see S3 Text). CI, confidence interval.
(TIF)

**S3 Fig. Risk perception by age group over time.** The raw proportion of Likert scale responses and self-reported risk status among adult participants.
(TIF)

**S4 Fig. Relative difference in mean contacts for adults by age group and study period with 95% CIs.** Relative differences calculated using a GAM with participants aged 18 to 39 as the reference period for each age group adjusted to the UK population by age and gender (when available) for the age groups 40 to 59 and 60+ years old. CI, confidence interval; GAM, generalised additive model.
(TIF)

**S5 Fig. Relative difference in mean contacts for children by age group and study period with 95% CIs.** Relative differences calculated using a GAM with participants aged 18 to 39 as the reference period for each age group adjusted to the UK population by age and gender (when available) for the age groups 0 to 4 and 5 to 17 years old. **(A)** Study periods in which schools were open. **(B)** Study periods in which schools were closed. CI, confidence interval; GAM, generalised additive model.
(TIF)

**S6 Fig. Contact matrices with absolute difference by time period. (A)** Contact matrices for all contacts in England for Lockdown 1, Lockdown 1 easing, and Relaxed restrictions (Diagonal) and the element-wise absolute difference between the matrices (off diagonal). Contacts censored to 50 contacts per participant. Lockdown 1 data from March 23 to June 3, 2020 and Lockdown 3 data from January 5 to 18, 2021. **(B)** Contact matrices for all contacts in England for Schools reopening, Lockdown 2 and Lockdown 2 easing (Diagonal), and the element-wise absolute difference between the matrices (off diagonal). Contacts censored to 50 contacts per participant. Lockdown 1 data from March 23 to June 3, 2020 and Lockdown 3 data from January 5 to 18, 2021. **(C)** Contact matrices for all contacts in England for Christmas, Lockdown 3 and Lockdown 3 easing (Diagonal), and the element-wise absolute difference between the matrices (off diagonal). Contacts censored to 50 contacts per participant. Lockdown 1 data from March 23 to June 3, 2020 and Lockdown 3 data from January 5 to 18, 2021.
(PDF)

**S1 Table. Number and percentage of participants in the adult panel who completed 1 round, 2 to 3 rounds, or 5 or more rounds, stratified by gender, age, country, NHS England region, and household size.** NHS, National Health Service.
(TIF)

**S2 Table. Number and percentage (by number of rounds completed) of parent participants who completed 1 round, 2 to 3 rounds, or 5 or more rounds, overall and stratified by gender, age, country, NHS England region, and household size.** Parents of children report gender by answering the question "As far as you know, which of the following describes how [NAME OF CHILD] thinks of themselves?," with the options "Male," "Female," "In another way," "Do not know," and "Prefer not to answer." NHS, National Health Service.
(TIF)

**S3 Table. Mean contacts over time by age with 95% CI of bootstrapped mean.** Mean reported contacts of participants weighted by age, gender, and weekday. CI, confidence interval.
(TIF)

**S4 Table. Mean number of reported contacts and number of participants in the POLY-MOD study [13] with 95% CI of bootstrapped mean.** Reported by age groups of 0 to 4, 5 to 17, 18 to 59, and 60 or more year, weighted by day of week. Participants were able to report the same contact in multiple settings. The number of participants in each age group is included. CI, confidence interval.
(TIF)

**S1 Text. CoMix contact survey questionnaires–final versions.** Final survey versions for panels A and B, C and D, and E and F.
(PDF)

**S2 Text. CoMix study proposal.** Original research proposal for the CoMix survey.
(PDF)

**S3 Text. Ipsos MORI survey recruitment and social categories.** Recruitment methods and social category mapped to reported occupations.
(PDF)

**S4 Text. STROBE statement.** Checklist of items that should be included in reports of cross-sectional studies. STROBE, STrengthening the Reporting of OBservational studies in Epidemiology.
(PDF)

**S1 Data. Mean contacts by age and date.** Bootstrapped mean contacts with 95% CIs. CI, confidence interval.
(XLSX)

## Acknowledgments

The authors would like to thank the team at Ipsos MORI, an international research agency, which implemented the surveys, with particular thanks to Sean Doherty, Alexandru Toreanik, Lorena Iovu, Corneliu Caloian, Alina Pancu, Rares Eremia, and Kim Brown for their work on coordinating and implementing multiple waves of the survey.

We would also like to thank the EpiPose management team and collaborating researchers, including Niel Hens, Jacco Wallinga, Philippe Beutels, Jantien Backer, James Wambua, Laurens Bogaardt, Veronika Jaeger, and Andre Karch.

The following authors were part of the CMMID COVID-19 working group. Each contributed in processing, cleaning and interpretation of data, interpreted findings, contributed to the manuscript, and approved the work for publication: Lloyd A C Chapman, Samuel Clifford, Thibaut Jombart, Kathleen O'Reilly, Jiayao Lei, Kaja Abbas, Fabienne Krauer, Stefan Flasche,

Alicia Rosello, Gwenan M Knight, Damien C Tully, Katherine E. Atkins, Rachael Pung, Rosalind M Eggo, David Hodgson, Mihaly Koltai, Yalda Jafari, Timothy W Russell, Frank G Sandmann, Oliver Brady, Naomi R Waterlow, Mark Jit, Fiona Yueqian Sun, Carl A B Pearson, William Waites, Emilie Finch, Akira Endo, Graham Medley, Ciara V McCarthy, Adam J Kucharski, Paul Mee, Hamish P Gibbs, Nicholas G. Davies, Billy J Quilty, Sophie R Meakin, C Julian Villabona-Arenas, Nikos I Bosse, Joel Hellewell, Simon R Procter, Yang Liu, Rachel Lowe, Rosanna C Barnard, Sam Abbott, Matthew Quaife, and Emily S Nightingale.

The following funding sources are acknowledged as providing funding for the working group authors. This research was partly funded by the Bill & Melinda Gates Foundation (INV-001754: MQ; INV-003174: JYL, MJ, YL; INV-016832: SRP; NTD Modelling Consortium OPP1184344: CABP, GFM; OPP1139859: BJQ; OPP1183986: ESN; OPP1191821: KO'R). BMGF (INV-016832; OPP1157270: KA). CADDE MR/S0195/1 & FAPESP 18/14389-0 (PM). EDCTP2 (RIA2020EF-2983-CSIGN: HPG). ERC Starting Grant (#757699: MQ). ERC (SG 757688: CJVA, KEA). This project has received funding from the European Union's Horizon 2020 research and innovation programme—project EpiPose (101003688: MJ, RCB, YL). FCDO/Wellcome Trust (Epidemic Preparedness Coronavirus research programme 221303/Z/20/Z: CABP). This research was partly funded by the Global Challenges Research Fund (GCRF) project "RECAP" managed through RCUK and ESRC (ES/P010873/1: TJ). HDR UK (MR/S003975/1: RME). HPRU (This research was partly funded by the National Institute for Health Research (NIHR) using UK aid from the UK government to support global health research. 200908: NIB, LACC). Innovation Fund (01VSF18015: FK). MRC (MR/N013638/1: EF, NRW; MR/V027956/1: WW). Nakajima Foundation (AE). NIHR (16/136/46: BJQ; 16/137/109: BJQ, FYS, MJ, YL; 1R01AI141534-01A1: DH; Health Protection Research Unit for Modelling Methodology HPRU-2012-10096: TJ; NIHR200908: AJK, RME; NIHR200929: CVM, FGS, MJ, NGD; PR-OD-1017-20002: AR). Royal Society (Dorothy Hodgkin Fellowship: RL). Singapore Ministry of Health (RP). UK DHSC/UK Aid/NIHR (PR-OD-1017-20001: HPG). UK MRC (MC_PC_19065—Covid 19: Understanding the dynamics and drivers of the COVID-19 epidemic using real-time outbreak analytics: NGD, RME, SC, TJ, YL; MR/P014658/1: GMK). Authors of this research receive funding from UK Public Health Rapid Support Team funded by the United Kingdom Department of Health and Social Care (TJ). UKRI (MR/V028456/1: YJ). Wellcome Trust (206250/Z/17/Z: AJK, TWR; 206471/Z/17/Z: OJB; 208812/Z/17/Z: SC, SFlasche; 210758/Z/18/Z: JH, SA, SRM; 221303/Z/20/Z: MK; UNS110424: FK). No funding for DCT.

## Disclaimers

The views expressed in this publication are those of the author(s) and not necessarily those of the NIHR or the UK Department of Health and Social Care.

## Author Contributions

**Conceptualization:** Amy Gimma, Kevin van Zandvoort, Kiesha Prem, Petra Klepac, G. James Rubin, W. John Edmunds, Christopher I. Jarvis.

**Data curation:** Amy Gimma, James D. Munday, Kerry L. M. Wong, Kevin van Zandvoort, Christopher I. Jarvis.

**Formal analysis:** Amy Gimma, James D. Munday, Kerry L. M. Wong, Kevin van Zandvoort, Christopher I. Jarvis.

**Funding acquisition:** Kevin van Zandvoort, W. John Edmunds, Christopher I. Jarvis.

**Investigation:** Amy Gimma, James D. Munday, Kerry L. M. Wong, Pietro Coletti, Kevin van Zandvoort, W. John Edmunds, Christopher I. Jarvis.

**Methodology:** Amy Gimma, James D. Munday, Kerry L. M. Wong, Pietro Coletti, Kevin van Zandvoort, Christopher I. Jarvis.

**Project administration:** Amy Gimma, Kevin van Zandvoort, W. John Edmunds, Christopher I. Jarvis.

**Resources:** Amy Gimma, Kevin van Zandvoort.

**Software:** Amy Gimma, James D. Munday, Kerry L. M. Wong, Kevin van Zandvoort, Christopher I. Jarvis.

**Supervision:** Petra Klepac, Sebastian Funk, W. John Edmunds, Christopher I. Jarvis.

**Validation:** Amy Gimma.

**Visualization:** Amy Gimma, Kerry L. M. Wong, Christopher I. Jarvis.

**Writing – original draft:** Amy Gimma, James D. Munday, Christopher I. Jarvis.

**Writing – review & editing:** Amy Gimma, James D. Munday, Kerry L. M. Wong, Pietro Coletti, Kevin van Zandvoort, Kiesha Prem, Petra Klepac, G. James Rubin, Sebastian Funk, W. John Edmunds, Christopher I. Jarvis.

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
