## [Editor Report · Decision Letter 0]

3 Jun 2021

Dear Dr Gimma, 

Thank you for submitting your manuscript entitled "CoMix: Changes in social contacts as measured by the contact survey during the COVID-19 pandemic in England between March 2020 and March 2021" for consideration by PLOS Medicine.

Your manuscript has now been evaluated by the PLOS Medicine editorial staff and I am writing to let you know that we would like to send your submission out for external peer review.

Please re-submit your manuscript within two working days, i.e. by Jun 07 2021 11:59PM.

Kind regards,

Caitlin Moyer, Ph.D.

Associate Editor

PLOS Medicine

---

## [Decision Letter · Decision Letter 1]

9 Aug 2021

Dear Dr. Gimma,

Thank you very much for submitting your manuscript "CoMix: Changes in social contacts as measured by the contact survey during the COVID-19 pandemic in England between March 2020 and March 2021" (PMEDICINE-D-21-02365R1) for consideration at PLOS Medicine. 

Your paper was evaluated by a senior editor and discussed among all the editors here. It was also discussed with an academic editor with relevant expertise, and sent to three independent reviewers, including a statistical reviewer. The reviews are appended at the bottom of this email and any accompanying reviewer attachments can be seen via the link below:

[LINK]

In light of these reviews, I am afraid that we will not be able to accept the manuscript for publication in the journal in its current form, but we would like to consider a revised version that addresses the reviewers' and editors' comments. Obviously we cannot make any decision about publication until we have seen the revised manuscript and your response, and we plan to seek re-review by one or more of the reviewers. 

We expect to receive your revised manuscript by Aug 30 2021 11:59PM. Please email us (plosmedicine@plos.org) if you have any questions or concerns.

We look forward to receiving your revised manuscript. 

Sincerely,

Caitlin Moyer, Ph.D.

Associate Editor 

PLOS Medicine

plosmedicine.org

From the Academic Editor:

1. Abstract: Abstract: Methods and Findings: The sentence “The mean of reported contacts among adults have reduced compared to previous surveys with adults aged 18 to 59 reporting a mean of 2.39 (95% CI 2.20 - 2.60) contacts to 4.93 (95% CI 4.65 - 5.19) contacts, and the mean contacts for school-age children was 3.07 (95% CI 2.89 - 3.27) to 15.11 (95% CI 13.87 - 16.41).” is confusing. Since previous surveys are mentioned, please provide some quantitative information on how they compare to these results.

2. Abstract: Conclusions: Line 37-38: “Recorded contacts reduced dramatically compared to pre-pandemic levels, with changes...” There is no data provided in the abstract on pre-pandemic levels and one must assume that the authors mean in reference to the Polymod study although this is not stated until much later in the paper.

3. Results: Contacts by Setting: Page 11: One would not expect household size to change due to Covid so it would be useful to compare the Polymod data on within household contacts to these results to gage how well the studies correspond to each other.

Other Editorial Points:

4. Title: Please revise your title according to PLOS Medicine's style. Your title must be nondeclarative and not a question. It should begin with main concept if possible. "Effect of" should be used only if causality can be inferred, i.e., for an RCT. Please place the study design ("A randomized controlled trial," "A retrospective study," "A modelling study," etc.) in the subtitle (ie, after a colon).

5. Data availability statement: PLOS Medicine requires that the de-identified data underlying the specific results in a published article be made available, without restrictions on access, in a public repository or as Supporting Information at the time of article publication, provided it is legal and ethical to do so. Please see the policy at

http://journals.plos.org/plosmedicine/s/data-availability

and FAQs at

http://journals.plos.org/plosmedicine/s/data-availability#loc-faqs-for-data-policy

Thank you for your willingness to make your data available on the zenodo platform. Please make it clear and provide the link(s) for the repository where the data will be made available. Please include any relevant data identifier, including DOI or accession number.

6. Author list: Please clarify if the individual members of the CMMID COVID-19 working group named in the Acknowledgements are intended to be named as individual authors or not. To qualify for authorship, all contributors must meet at least one of the seven core contributions (conceptualization, methodology, software, validation, formal analysis, investigation, data curation), as well as at least one of the writing contributions (original draft preparation, review and editing). Please see our policy on group authorship here: https://journals.plos.org/plosmedicine/s/authorship#loc-group-authorship

7. Abstract: Please combine the Methods and Findings sections into one section, “Methods and findings”.

8. Abstract: Background: Please provide more context supporting the advance and importance of the study. The final sentence should clearly state the study question.

9. Abstract: Methods and Findings: Please include the study design, population and setting, number of participants, and main outcome measures early on in the description of the Methods.

10. Abstract: Conclusions: Please address the study implications without overreaching what can be concluded from the data; the phrase "In this study, we observed ..." may be useful.

11. Author Summary: At this stage, we ask that you include a short, non-technical Author Summary of your research to make findings accessible to a wide audience that includes both scientists and non-scientists. The Author Summary should immediately follow the Abstract in your revised manuscript. This text is subject to editorial change and should be distinct from the scientific abstract. Please see our author guidelines for more information: https://journals.plos.org/plosmedicine/s/revising-your-manuscript#loc-author-summary

12. Background: Please rename this section as “Introduction” and please expand the Introduction to fully address past research. Please clearly emphasize the novelty, the need for and potential importance of your study. Please conclude the Introduction with a clear description of the study question or hypothesis.

13. Methods: Please ensure that the study is reported according to the STROBE guideline, or the most appropriate reporting guideline for your study, and include the completed STROBE or other checklist as Supporting Information.

Please add the following statement, or similar, to the Methods: "This study is reported as per the Strengthening the Reporting of Observational Studies in Epidemiology (STROBE) guideline (S1 Checklist)."

14. Methods: Did your study have a prospective protocol or analysis plan? Please state this (either way) early in the Methods section. Thank you for including the Protocol as a supporting information file. Please note if this was prospectively developed, and please make sure that the Methods section transparently describes when analyses were planned, and when/why any data-driven changes to analyses took place. Any changes in the analysis-- including those made in response to peer review comments-- should be identified as such in the Methods section of the paper, with rationale.

15. Methods: Please specify the nature of informed participant consent. Please clarify if the participants were compensated to participate (this was mentioned in the protocol).

16. Methods: Line 160: Please clarify if these are national hospitalization data for England.

17. Methods: Please describe how “social grade” was calculated/determined from the survey response.

18. Results: Mean contacts: If possible please present the mean daily contacts for study periods in table format, stratified by setting, perceived risk, similar to results presented in Table S3.

19. : Line 269 and line 356: We suggest clarifying “during the Christmas period” with the months/dates of the time period instead.

20. Discussion: Please present and organize the Discussion as follows: a short, clear summary of the article's findings; what the study adds to existing research and where and why the results may differ from previous research; strengths and limitations of the study; implications and next steps for research, clinical practice, and/or public policy; one-paragraph conclusion. It would be helpful to expand the discussion of the pre-pandemic patterns, to highlight the novelty, advance, and policy implications of the findings presented here and also to provide support for your conclusion that contacts are lower than “normal levels” reported in previous years.

21. Discussion: Line 389: Please avoid language that implies a causal relationship: “Although easing of restrictions did lead to an increase in contacts, this did not necessarily occur immediately.”

22. Discussion: Line 421-423: Please clarify and reference the mobility indices mentioned here: “A number of mobility indices such as google and facebook have been made available during the pandemic, which also provide an indication of…” and we suggest capitalizing Google and Facebook.

23. Table 1: It may be helpful, in the Methods or Results, or as supporting information, to have some information on the general characteristics of each lockdown relevant to social contact, in terms of restrictions (for example, what is meant by “relaxed” restrictions, “easing” of restrictions, etc.).

24. Table 2: Please provide summary demographic data including part time employment status, perceived risk, income level, and precaution information.

25. Figure 2: It would be helpful to have dates included associated with each lockdown-related study period.

26. Table 3: Were any statistical tests done to evaluate changes in relative differences in contacts among age or other groups across study periods?

27. Table S1 and S2: Please clarify if “day of week” is indicated in the table.

Comments from the reviewers:

Reviewer #1: This paper estimates contact patterns over a large period of time during the Covid-19 epidemic in UK. They describe overall changes with respect to pre-pandemic levels and changes between March 2020 and March 2021 due to governmental interventions during the epidemic. They provide a unique dataset on the evolution of contact patterns during the COVID-19 pandemic that can be used to inform future outbreak response and can be applied to transmission of other infectious diseases in similar settings.

Overall, the article is well written, it addresses an important area of research, and contains a lot of work. Considering the need for studies that quantify behavioural changes during a pandemic, this study could have an important contribution to science. 

I have some minor points on some part of the Results and the Discussion section.

Pag12, lines 299-306: in its current form the paragraph is confused, and it is not very clear to what they refer by relative differences. Relative with respect to what?

Reading the manuscript and with the mounting scientific evidence on the difference in the contribution of indoor/outdoor contacts to SARS-Cov2 transmission, it is quite startling not to see any reference to indoor/outdoor contacts throughout the results and any table reporting contacts by setting even in the supplementary material. If you registered this information, I suggest you discuss it in the paper. Otherwise, I would state this as a quite important limitation of the study and important data to collect in future studies.

Finally, I would discuss better your findings with respect to pre-pandemic observed contact patterns, both in terms of mean number of contacts per person and age specific patterns. 

Reviewer #2: This study provides a descriptive analyses showing the mean number of contacts people reported and how these differed during three national lockdowns, periods with more relaxed restrictions, and over the Christmas holiday period. 

The study design is well thought through and comprehensive, and the technically appropriate methodology has been clearly and concisely communicated. 

The results are fairly presented and accurately interpreted, and the main study limitations have been thoroughly explored in the discussion section.

Reviewer #3: PMEDICINE-D-21-02365R1

Thank you for the opportunity to review the article entitled "CoMix: Changes in social contacts as measured by the 1 contact survey during the COVID-19 pandemic in England between March 2020 and March 2021" (PMEDICINE-D-21-02365R1). This paper provides some extremely useful and relevant information about contact patterns during the past year. This data should be disseminated and the authors have done an excellent and thorough job of explaining the study and presenting the data. However, more context is needed. Right now this reads as a data report, when it is my understanding that the readers of PLOS Medicine will be interested in some context. Perhaps one or two more paragraphs on why this is important and why this data is being reported might help. Some contextual literature on adherence and the importance of social distancing in this pandemic and past pandemics might be relevant (Williams et al., 2020a,b). Also, the main message of the paper, that largely people are following government guidelines and policy is the main driver of contacts, is somewhat lost in the methodology and data. I would suggest more of a 'why we care' in the introduction and some emphasis on the main message of the findings in the discussion and abstract prior to publication. That being said, the methodology and results were of an extremely high quality and the CoMix study is excellent. However, readers may not be as knowledgeable about this type of methodology, so clearer contextualization is necessary.

[LINK]

---

## [Decision Letter · Decision Letter 2]

13 Dec 2021

Dear Dr. Gimma,

Thank you very much for re-submitting your manuscript "Changes in social contacts as measured by the CoMix contact survey during the COVID-19 pandemic in England between March 2020 and March 2021: a repeated cross-sectional study" (PMEDICINE-D-21-02365R2) for review by PLOS Medicine.

I have discussed the paper with my colleagues and the academic editor and it was also seen again by one of the reviewers. I am pleased to say that provided the remaining editorial and production issues are dealt with we are planning to accept the paper for publication in the journal.

[LINK]

We look forward to receiving the revised manuscript by Dec 20 2021 11:59PM.   

Sincerely,

Caitlin Moyer, Ph.D.

Associate Editor 

PLOS Medicine

plosmedicine.org

Requests from Editors:

1. Title: We suggest revising slightly: “Changes in social contacts in England during the COVID-19 pandemic between March 2020 and March 2021 as measured by the CoMix contact survey: A repeated cross-sectional study”

2. Data availability statement: Thank you for your willingness to share your data. At this time please revise this statement to indicate that all data and code underlying the study is accessible from the three links provided and/or provide all links necessary for interested parties to access the data.

3. Authors: Please include CMMID COVID-19 working group in the author list, and list the individual members of the CMMID COVID-19 working group in the Acknowledgements.

4. Abstract: Line 19: Please consider changing “ lower restrictions” to “fewer restrictions” or similar.

5. Abstract: Methods and Findings: Please provide more detail on how the survey of the population was implemented (e.g. internet based banner and social media ads, email campaigns).

6. Abstract: Line 31-32: We suggest revising to: “Changes in contact patterns were observed over time, and associated with participants’ age, personal risk factors, and perception of risk.”

7. Abstract: Line 35-37: Please clarify if this is for adults or children, as contact findings for both age groups were presented above: “This demonstrates a sustained decrease in social contacts compared to a mean of 11.08 (95%CI 10.54 to 11.57) contacts per participant overall as measured by the POLYMOD social contact study in 2005/06.”

8. Abstract: Methods and Findings: In the last sentence of the Abstract Methods and Findings section, please describe the main limitation(s) of the study's methodology.

9. Abstract: We suggest re-organizing and consolidating the Conclusions section, slightly. We suggest moving the first sentence to the end of the Conclusions, and combining it with the final sentence.

10. Abstract: Conclusions: At line 42-43: We suggest revising the sentence to mention the setting: “...the CoMix survey provides a unique repeated cross-sectional data set for the year subsequent to the first lockdown in England…”

11. Author summary: We suggest consolidating and reducing the length of each point of the summary. Please consistently use past tense in the section “What did the researchers do and find?”

12. Author summary: Why was this study done? Please use “COVID-19” consistently throughout. We suggest revising the first and second points to:

-“Mathematical models can be used to better understand the transmission dynamics of COVID-19, and could be strengthened by empirical evidence of the number of social contacts made under pandemic conditions.”

- “There is a need for real-time social contact data to inform these models, because we predict that behavior will change due to perceived risk and in response to government policies restricting social contact of the course of the panemic, such as during lockdowns.”

13. Author summary: What do these findings mean? For the first bullet point, please summarize the implications of the observed decreases in social contact over the first year of the pandemic. We suggest shortening and combining the first three existing bullet points into one point.

14. Author summary: We suggest revising the fourth bullet point to “These data may be used by researchers and policymakers to monitor changes…”

15. Introduction: Line 127-135: We suggest that this paragraph may be more suited to the Discussion/Conclusions than the Introduction.

16. Methods: Ethics statement: Please indicate the nature of informed participant consent. Thank you for indicating that participants agreed to participate, and awareness of right to decline to participate, but please explicitly note participant informed consent to participate, and whether this consent was requested at each survey round.

17. Methods: Please include the completed STROBE checklist as Supporting Information. Please add the following statement, or similar, to the Methods: "This study is reported as per the Strengthening the Reporting of Observational Studies in Epidemiology (STROBE) guideline (S1 Checklist)."

18. Methods: Prospective analysis plan: Thank you for your response indicating some changes made to your original study protocol. Please explicitly make note of this in the text, where applicable.

19. Results: Line 372-377: If possible it might be helpful to also provide the non-lockdown mean recorded contacts for 0-4 year old children, and also the lockdown mean contacts for 5-17 year old children. This make make the comparisons with the POLYMOD study easier to put into context.

20. Results: Line 381-413 and 435-453: It may be helpful to illustrate these observations with some quantitative data.

21. Discussion: Line 494: Please remove the heading “Summary of Findings”

22. Discussion: Line 504-506: Please revise the wording here to avoid implications of causality (e.g. the use of the word determinant): “The survey results suggest that government action was a major factor in the mean number of social contacts, with contact rates dropping markedly during every lockdown. However, it was not the only determinant.”

23. Discussion: Line 524: Please avoid the term “significantly” here, unless describing a statistical difference.

24. Discussion: Line 551: Please consider if “longitudinal” is the best term to describe the survey, or if repeated cross-sectional (or similar) would be accurate.

25. Discussion: Line 554: Please revise to: “ A number of mobility indices such as those implemented by Google and Facebook…”

26. Discussion: Please be sure to organize the Discussion as follows: a short, clear summary of the article's findings; what the study adds to existing research and where and why the results may differ from previous research; the strengths and limitations of the study; the implications and next steps for research, clinical practice, and/or public policy; one-paragraph conclusion. The discussion of the limitations seems as if it should come after the “CoMix in context” section at line 542. Also, please include a paragraph describing implications for research, practice and policy before the Conclusion.

27. Figure 6: In the legend, it would be helpful to summarize slightly the key take home point of the contact matrices for each period.

28. Table 3: Please provide the mean contacts for each age group during each period in a table, in addition to the relative differences (similar to Table S3, but for each of the 9 periods). Also, please note if statistical tests were done to compare differences in contacts during different periods within an age group.

29. Page 34: Please remove the section “Ethics approval and consent to participate” as this is already included in the Methods section (please be sure all information is complete and accurate). Please remove the sections Consent for Publication, Availability of Data and Materials, Author’s Contributions, Competing Interests, and Funding. Please be sure all information is entered completely in the relevant sections of the manuscript submission system.

30. Acknowledgments: Please list the Centre for Mathematical Modelling of Infectious Disease Working Group Authors in the Acknowledgements.

31. References: Please double check formatting of each reference. Please remove the underline for reference 30. Please use the "Vancouver" style for reference formatting, and see our website for other reference guidelines https://journals.plos.org/plosmedicine/s/submission-guidelines#loc-references

32. Supporting Information files: Please be sure that each supporting information item has a number and name description. You may use almost any description as the item name of your supporting information as long as it contains an "S" and number. For example, “S1 Appendix” and “S2 Appendix,” “S1 Table” and “S2 Table,” and so forth. Match the names of your supporting information files with the supporting information captions within your manuscript. For example, a PDF file for “S2 Fig.” must be named “S2_fig.pdf”. In the published article, supporting information files are accessed only through a hyperlink attached to the captions. For this reason, you must list captions at the end of your manuscript file. You may include a caption within the supporting information file itself, as long as that caption is also provided in the manuscript file. Do not submit a separate caption file.

33. Supporting Information: Contact Matrices: If possible, please provide brief description of the matrices that facilities understanding by non-specialists who may not be familiar with the interpretation of matrices.

34. Table S4: Please reference the POLYMOD study in the legend.

35. Figure S1: Please increase the font size used, if possible.

36. Figure S2: Please explain “ABC1” and “C2DE” in the legend.

37. Figure S3: Please provide a label for the y axis, if possible.

Comments from Reviewers:

Reviewer #3: The authors have addressed the comments comprehensively and I have no further additions/changes.

[LINK]

---

## [Editor Report · Decision Letter 3]

6 Jan 2022

Dear Dr Gimma, 

On behalf of my colleagues and the Academic Editor, Megan Murray, I am pleased to inform you that we have agreed to publish your manuscript "Changes in social contacts in England during the COVID-19 pandemic between March 2020 and March 2021 as measured by the CoMix survey: A repeated cross-sectional study" (PMEDICINE-D-21-02365R3) in PLOS Medicine.

Please also address the following editorial points:

-Author byline: Including the CMMID Working Group as a group author in the byline as you have done is acceptable. Similar to the prior publication you mentioned (https://doi.org/10.1371/journal.pmed.1003815) you may note that “Membership of CMMID COVID-19 Working Group is provided in the Acknowledgements.” The list of members of the group (Centre for Mathematical Modelling of Infectious Disease Working Group Authors) was not found in the current version of the manuscript, but may be included in the Acknowledgements. 

-Data availability: Please also provide the GitHub link for the analysis code, as indicated on line 247 of the Methods (e.g. https://github.com/amygimma/comix_uk_summary_analysis)

-Results: Line 501-505: Based on Figure 3C, please check if this should be revised to: “The highest mean number of contacts for those who did not report that COVID-19 would be a serious illness for them was 5.65 (95% CI 5.12 to 6.20) in mid-August 2020 compared to 4.62 (95% CI 4.20 to 5.04) for those who did agree that COVID-19 would be a serious illness for them in the same time period.”

-Discussion: Line 608: Please qualify with “To the best of our knowledge, CoMix is the largest…”

-Reference 9: Please update the citation (doi: 10.1097/EDE.0000000000001412).

-Reference 13: Please change to “PLoS Med. 2008 Mar 25;5(3):e74.”

-Reference 28: Please update the citation (https://doi.org/10.1186/s12916-021-01924-7)

-Reference 29: Please update the citation (doi: 10.1098/rstb.2020.0283)

-STROBE Checklist: Thank you for including the completed checklist. Please revise the checklist, using section and paragraph numbers to refer to locations within the text (e.g. Methods, paragraph 1). Please remove the line numbers (please do not use line or page numbers).

PRESS

Sincerely, 

Caitlin Moyer, Ph.D. 

Associate Editor 

PLOS Medicine